

# Variations in the physicochemical and optical properties of natural aerosols in Puerto Rico - Implications for climate

Héctor Rivera[1], John A. Ogren[2,3], Elisabeth Andrews[3], Olga L. Mayol-Bracero[4]

[1]Department of Physics, University of Puerto Rico - Rio Piedras, San Juan, Puerto Rico
[2]Earth Systems Research Laboratory, National Oceanic and Atmospheric Administration, Boulder, Colorado
[3]Cooperative Institute for Research in Environmental Sciences, University of Colorado, Boulder, Colorado
[4]Department of Environmental Sciences, University of Puerto Rico - Rio Piedras, San Juan, Puerto Rico

*Correspondence to*: Olga L. Mayol-Bracero (omayol@ites.upr.edu)

**Abstract.** Since 2005, we have monitored the physicochemical and optical properties of aerosols at the Cape San Juan Atmospheric Observatory, Puerto Rico. Based on the Hybrid Single-Particle Lagrangian Integrated trajectories (HYSPLIT) and satellite imagery from the Volcanic Ash Advisory Center (VAAC) in Washington D.C., Moderate Resolution Imaging Spectroradiometer (MODIS), and Saharan air layer (SAL) images, we grouped natural aerosols in three categories: marine, African dust and volcanic ash. A sun-sky radiometer from the NASA's AErosol RObotic NETwork (AERONET) assessed the total aerosol optical depth and its fine fraction. A 3-wavelength nephelometer and particle soot absorption photometer assessed the scattering and absorption coefficients. Two impactors segregated the submicron ($D_p < 1$ µm) particles from the total ($D_p < 10$ µm) enabling us to calculate the sub-micron scattering and absorption fractions. The measured variables served to calculate the single scattering albedo and radiative forcing efficiency. All variables except the single scattering albedo making up the aerosol climatology for Puerto Rico had different means as function of the aerosol category at $p<0.05$. For the period 2005-2010, the largest means ± 95% confidence interval of the scattering coefficient ($53 \pm 4$ Mm$^{-1}$), absorption coefficient ($1.8 \pm 0.16$ Mm$^{-1}$), and optical depth ($0.29 \pm 0.03$), suggested African dust is the main contributor to the columnar and surface aerosol loading in summer. About two thirds (63%) of the absorption in African dust was due to the coarse mode and about one third due to the fine mode. In volcanic ash, fine aerosols contributed 60% of the absorption while coarse contributed 40%. Overall, the coarse and fine modes accounted for ~80% and 20% of the total scattering. The African dust load was 3.5 times the load of clean marine, 1.9 times greater than the clean marine with higher sea salt content, and 1.7 times greater than volcanic ash. African dust caused 50% more cooling than that volcanic ash at the top of the atmosphere and 50% more heating than that of volcanic ash within the marine boundary layer (MBL).



## 1 Introduction

Atmospheric aerosols present high uncertainty in climate prediction (Boucher et al., 2013) because of their differences in amount, in size, and in index of refraction, which in turn depends on the chemical composition and on the source. Different physicochemical properties of aerosols result in diverse optical properties influencing climate and environment
in many ways (Seinfeld and Pandis, 1998). Ogren (1995) pointed out that we need to evaluate the aerosols climate-forcing properties to know their spatial distribution, their physical, their optical, and their cloud-nucleating properties. We also need to evaluate suitable radiative transfer models, and cloud physics. To contribute with this need, we evaluated the climate-forcing properties of natural aerosols in Puerto Rico.

Extensive research exists on the topic of anthropogenic aerosol climate forcing (Boucher et al., 2013); however, little
has been done regarding natural aerosols in the Caribbean. The Caribbean is exposed to different natural aerosols such as those coming from marine sources, mineral dust from Africa and volcanic ash from the Soufriere Hills volcano in the island of Montserrat (e.g., Li-Jones and Prospero, 1998; Gioda et al., 2011; Prospero and Mayol-Bracero, 2013; Valle-Diaz et al., 2016; Wex et al., 2016). These three types of aerosols are the focus of this study.

For marine aerosols, oceans produce the greatest primary aerosol mass emissions (Warneck 1988) and are a key source of secondary atmospheric aerosols (O'Dowd and Smith, 1993; O'Dowd et al., 2004). The mass concentration and size distribution of marine aerosols depend on the wind speed (Woodcock, 1953; Lovett, 1978; Blanchard and Woodcock, 1980). Wind speed correlates directly with the sea-salt amount, but wind speed only explains part of the variance (Quinn and Coffman, 1999; Smirnov et al., 2003). Sea-salt is non-absorbing and comprises much of the MBL aerosol mass,
changing the radiative balance through scattering of visible light (Quinn et al., 1996; Winter and Chýlek, 1997). The sub-micron non-sea salt (nss) sulfate from biological activity scatters visible light efficiently also serving as cloud condensation nuclei (Charlson et al., 1991; Jacobson, 2001).

Turning to African dust, hundreds of teragrams of it reach the atmosphere every year (Huneuus et al., 2011), with large
variations in emissions, in space and in time (Prospero, 1999; Vinoj et al., 2004). These pace-time changing amounts result in poorly characterized African dust radiative-forcing properties (Liao and Seinfeld, 1998; Sokolik and Toon, 1999; Sokolik et al., 2001, which in turn reduces the accuracy of numerical models for predicting climate change (Houghton et al., 2001). In addition, African dust changes the radiative balance scattering and absorbing solar and terrestrial radiation. The dominant absorbing species in African dust are the iron oxides (Sokolik and Toon, 1999;
Moosmüller et al., 2009).

Such as African dust, volcanic aerosols change the Earth's radiative balance by scattering and by absorbing solar radiation (Bohren and Huffman, 1999) because volcanic aerosols can hold volcanic ash that in turn holds iron oxides that absorb sunlight (Seinfeld and Pandis, 1998; Kokhanovsky, 2008). Volcanic ash is hard, abrasive, and acidic
(Krotkov et al., 1999; Housley et al., 2002) resulting in an aviation hazard. Evidence has shown engine failure in aircrafts flying through volcanic ash (Krotkov et al., 1999; USGS Fact Sheet, 2006).

Aerosols studies in the Caribbean goes back to the 1970s when Prospero et al. (1970) and Prospero and Carlson (1972) highlighted that synoptic outbreaks of Saharan dust that occur from late spring to fall. These outbreaks extend from
western Africa across the tropical Atlantic to the Caribbean. Other studies of transported African dust in the Caribbean,





particularly in Puerto Rico, include those of Reid et al. (2002), Gioda et al. (2011), Fitzgerald et al. (2015), Spiegel et al. (2014); Denjean et al. (2016), Raga et al. (2016) and Valle-Diaz et al. (2016) as well as observations employing active remote sensing instruments such as Lidars (Burton et al., 2015). From these studies, we highlight the Puerto Rico Dust Experiment (PRIDE) (Reid et al., 2003a, b), the only study that included the radiative, microphysical, and transport

properties of African dust. Reid et al. (2002) reported that during the first half of PRIDE (June 2000), dust had the highest concentrations in the marine and convective boundary layers, with lower dust concentrations above the trade inversion despite a strong Saharan Air Layer (SAL). PRIDE showed that coarse marine aerosols produced most of the scattering and optical depth in Puerto Rico. However, the African dust coarse mode generates most of the optical depth in spring and summer (Reid et al., 2003b). Also, with single-particle analyses E. A. Reid et al. (2003) reported that

elemental iron composes ~2.5-3% of the total dust mass (assuming aluminum is 8% of the total mass). PRIDE excluded characterizing volcanic ash and characterizing the long-term variability of the climate-forcing properties of aerosols in Puerto Rico.

In this article, we characterize the climate-forcing properties of natural aerosols in Puerto Rico. We analyze aerosol data

collected in Puerto Rico from 2005-2010 to: 1) classify local aerosols by source (marine, African dust and volcanic ash), 2) characterize means and variabilities in climate-forcing properties of aerosols from these three natural sources and report the monthly climatology of aerosols in Puerto Rico, 3) test the hypothesis that "means and variability of aerosols from different sources differ significantly at $p<0.05$", and 4) determine if we can distinguish the kind of aerosol only knowing their mean and variability in the direct climate-forcing properties.

**2 Experimental**

The sampling site was the Cape San Juan Atmospheric Observatory, at the natural reserve of Cabezas de San Juan, Puerto Rico (CPR), with coordinates (18°22.85'N, 65°37.07'W), managed by the Atmospheric Chemistry and Aerosols Research group at the University of Puerto Rico – Rio Piedras Campus, and supported by the Aerosol Group of the Global Monitoring Division at the National Oceanic and Atmospheric Administration's Earth System Research

Laboratory (NOAA/ESRL). CPR is a coastal site influenced by the trade winds most of the year where the absence of large land areas upwind reduces anthropogenic aerosols. African dust disturbs the marine environment of CPR from late spring to mid fall with stronger events in late spring and summer. Emissions from the Soufriere Hills volcano in Montserrat also disturb CPR if the low-level winds are southeast because Puerto Rico is about 400 km northwest of Montserrat.


**2.1 In-situ aerosol measurements**

Delene and Ogren (2002) have described the aerosol monitoring. The TSI model 3563 integrating nephelometer measures the aerosol scattering ($\sigma_{sp}$) and backscattering ($\sigma_{bsp}$) coefficients at 450, 550 and 700 nm. In addition, a

Radiance Research Particle Soot Absorption Photometer (PSAP) measures the aerosol absorption coefficient ($\sigma_{ap}$) at 467, 530, and 660 nm. The absorption at 530 nm was adjusted by log-log interpolation to 550 nm to yield $\sigma_{ap}$ and $\sigma_{sp}$ at the same wavelength (550 nm). Upstream of the nephelometer and PSAP a heater warms the aerosol sample to reduce the relative humidity to values around 40%. Two switched impactors segregate the aerosols in sub-micron ($D_p < 1$ µm) and total ($D_p < 10$ µm) fractions to compare the fine with the total aerosol contributions.






### 2.1.1 Description of in-situ variables

Variables $\sigma_{sp}$, $\sigma_{ap}$ and extinction coefficient ($\sigma_{ext}$) are extensive parameters (Ogren, 1995). Extensive parameters depend on aerosol amount and are additive. We report $\sigma_{sp}$ and $\sigma_{ap}$ in Mm$^{-1}$ (1 Mm$^{-1}$ = 10$^{-6}$ m$^{-1}$).

Calculating the sub-micron scattering ($R_{sp}$) and absorption ($R_{ap}$) fractions allowed us to test the contribution of the submicron mode to the total scattering or to the total absorption. The $R_{sp}$ and $R_{ap}$ are intensive (i.e., independent of aerosol amount) and non-dimensional parameters associated with the scattering or absorbing particles size distributions (Ogren 1995). Intensive and extensive parameters are variables in chemical transport and radiative transfer models. We calculated $R_{sp}$ and $R_{ap}$ with Eqns. (1) and (2) from Delene and Ogren (2002).

$$R_{sp}(D_p) = \frac{\sigma_{sp}(D_p < 1\mu m)}{\sigma_{sp}(D_p < 10\mu m)} \qquad (1)$$

$$R_{ap}(D_p) = \frac{\sigma_{ap}(D_p < 1\mu m)}{\sigma_{ap}(D_p < 10\mu m)} \qquad (2)$$

The scattering Ångström exponent (å) is an intensive parameter describing the spectral dependence of the scattering (Eqn. (3))

$$\sigma_{sp} = Ɓ\lambda^{-å} \qquad (3)$$

with $\sigma_{sp}$ the scattering coefficient, Ɓ the scattering coefficient at a wavelength λ of one μm, and å the scattering Ångström exponent. We calculated the scattering Ångström exponent with Eqn. (4).

$$å = -\frac{\log(\sigma_{sp}^{550}/\sigma_{sp}^{700})}{\log\left(\frac{550}{700}\right)} \qquad (4)$$

In Eqn. (4), $\sigma_{sp}^{550}$ and $\sigma_{sp}^{700}$ are the scattering coefficients at 550 and 700 nm. The scattering Ångström exponent qualitatively measures the sizes of scattering particles. The scattering Ångström exponent å varies with the particle size distribution such that lower Ångström exponents are due to distributions dominated by larger scattering particles, and vice versa.

In addition, we calculated the single scattering albedo ($\omega_0$), a non-dimensional intensive quantity, to estimate the contribution of scattering to extinction with Eqn. (5).

$$\omega_0 = \frac{\sigma_{sp}}{\sigma_{ap} + \sigma_{sp}} \qquad (5)$$

The $\omega_0$ is part of the radiative forcing per unit of optical depth (ΔRF/AOD) at the top of the atmosphere, called radiative forcing efficiency after Sheridan and Ogren (1999). The ΔRF/AOD depends on the aerosol size through the upscatter fraction β, the scattering and absorption through $\omega_0$, and on seven geophysical quantities. We calculated the upscatter fraction with Eqn. β = 0.0817 + 1.8495b - 2.9682b$^2$, where the backscatter fraction b was calculated with Eqn. b =





$\sigma_{bsp}/\sigma_{sp}$. This parameterization, presented by Sheridan and Ogren (1999), omits the dependence of $\beta$ with the zenith angle. The radiative forcing efficiency for daytime-average was calculated with Eqn. (6).

$$\frac{\Delta RF}{AOD} = -DS_0 T_{at}^2 (1 - A_c)\omega_0\beta\left\{(1 - R_s)^2 - \left(2\frac{R_s}{\beta}\left[\left(\frac{1}{\omega_0}\right) - 1\right]\right)\right\} \qquad (6)$$

Eqn. (6) assumes a constant geographical surface reflectance ($R_s$) and atmospheric transmission ($T_{at}$), with the values for the fractional day length (D) = 0.5, solar constant ($S_o$) = 1370 W m$^{-2}$, $T_{at}$ = 0.76, fractional cloud amount ($A_c$) = 0.6, and $R_s$ = 0.15 such as proposed by Haywood and Shine (1995). The aerosol measurements are at 550 nm (Delene and Ogren, 2002).

**2.2 Ground-based remote aerosol measurements**

A CIMEL Electronique 318A spectral radiometer, part of NASA's AErosol RObotic NETwork (AERONET) Cape_San_Juan (CPR) station, assessed the total aerosol optical depth (AOD) and aerosol optical depth fine fraction
(AODFF). Holben et al. (1998) detail how to find AOD. The total aerosol optical depth AOD is the integral in the vertical of $\sigma_{ext, \lambda}$, AOD is a non-dimensional extensive parameter. AERONET finds the fine fraction with the Spectral Deconvolution Algorithm (SDA; O'Neill, 2001, 2003). Also, AERONET uses the number size distributions to derive the volume size distribution through an inversion algorithm by Dubovik and King (2000). AERONET columnar measures include AOD at 1020, 870, 675, 500, 440, 380, and 340 nm. We used AOD at 500 nm to compare with the $\sigma_{sp}$
at 550 nm.

**2.3 Models and satellite data**

We used the Hybrid Single Particle Lagrangian Integrated Trajectory (HYSPLIT) Model (Draxler and Hess, 1998;
Draxler and Rolph, 2013) to estimate trajectories of aerosols from the sources to CPR. Moderate Resolution Imaging Spectroradiometer (MODIS) images served to sense possible African dust, volcanic emissions, or clean marine. SAL images proved useful to sense possible African dust over CPR. We used SAL images from http://tropic.ssec.wisc.edu/archive/. Images from the Volcanic Ash Advisory Center (VAAC) in Washington D.C. served to sense volcanic ash if MODIS images were unavailable. VAAC obtains data from three Geostationary Operational
Environmental Satellite (GOES) satellites (GOES-11, 10 GOES-12, and GOES-13) covering from the central Pacific to the eastern Atlantic.

**2.4 Data processing and quality control**

Nephelometer data were corrected for instrumental non-idealities, such as truncation error, after Anderson and Ogren (1998). The method corrects scattering measurements over integration angles of ~7°–170° and ~90°–170° to the full 0°–180° and 90°–180° ranges, based on the measured scattering Ångström exponent (å). The scattering Ångström exponent qualitatively describes the scattering particles sizes (Anderson and Ogren, 1998). Concerning particles larger than the scattered light wavelength, the truncation error is greater than that of other systematic errors and can become greater by
a factor of two. Uncertainties also arise because the nephelometer is calibrated with a gas (Carbon dioxide), which scatters in a Rayleigh regime, and aerosols scatter in the Mie regime. Scattering from gas molecules (Rayleigh scattering)



is subtracted from the total scattering to find the scattering by aerosol particles. Corrections to PSAP data were based on Bond et al. (1999), reporting a 20-30% overall overestimation of absorption measured by the PSAP due to scattering aerosols on the filter before applying correction. Ogren (2010) extended the Bond et al. (1999) corrections to apply to measurements of a 3-wavelength PSAP. We manually edited one-minute data from the nephelometer and PSAP to

invalidate bad data associated with equipment maintenance and malfunction. The edited 1-minute data files were averaged to create the daily averaged data files. In addition, we checked for consistency of data and impossible values such as $R_{sp}$, $\omega_o > 1$. We kept these data in the database but omitted them from the plots. In our dataset the measurements of absorption and its related variables began in 2006. We used AERONET level 2 data, screened for clouds and quality assured by NASA (Holben et al., 2006).

**3 Results and Discussion**

First, we describe how we assigned the aerosol categories and a few exceptions to the method. Second, we analyze means and variations in the climate-forcing properties, discussing how the differences in the means and the variations among aerosol categories support or contradict our hypothesis. Finally, we discuss the monthly variation of the climate-forcing properties. For all aerosol categories the trajectories chosen were at 100, 500 and 1000 m in height.

**3.1 Aerosol Classification by Source**

We present our criteria to label aerosol categories in Table 1, and the sequence of steps followed is presented as a flowchart in Table 2. The method, however, produces uncertainty because we classified average categories in a 24h-

basis, but MODIS images are unavailable at night.

**3.1.1 Clean marine aerosols**

Clean marine (CM) aerosols form in the ocean with imperceptible influence from other categories, natural or

anthropogenic, with $\sigma_{sp} \leq 20$ Mm$^{-1}$, $\sigma_{ap} \leq 0.6$ Mm$^{-1}$, AOD $\leq 0.1$, and trajectories only over the ocean. Trajectories were at 06 and 12Z for 100, 500 and 1000 m above sea level. The clean marine with greater sea salt content (CMS) aerosol met the criteria in CM but, $\sigma_{sp} > 20$ Mm$^{-1}$ and AOD $> 0.1$. This definition was based on reports by Kleefeld et al. (2002) on the dependence $\sigma_{sp}$ with wind speed (square of the speed) and Lewis and Schwartz (2004) that wind speed is a main driver producing natural marine aerosols. VAAC and MODIS images served to verify non-marine aerosols.


**3.1.2 African dust**

To find African dust (AD), we searched MODIS images for dust clouds leaving western Africa in spring, summer, and fall (see Table 1). If we saw elevated aerosol loads in Puerto Rico about 6-7 days later, we classified as African dust. An

example is the dust cloud over Dakar in May 28, 2010 (Figure 1a), seen over Puerto Rico on June 3, 2010 (Figure 1b). AD in Figures 2a and 2b was, as in other AD events, light brown covering large areas of the Caribbean and Atlantic. AD trajectories were from the east, east-southeast, or southeast. Strong AD episodes such as the event on June 3, 2010 were easily identifiable with MODIS.


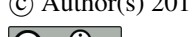



### 3.1.3 Volcanic aerosols

We assigned volcanic aerosols (VA) if the Soufriere Hills' volcano emitted simultaneously with southeast low-level winds, trajectories, and cloud streaks orientation (see Table 1). Figure 2 shows an example during January 9, 2007. We also used cloud streak orientations to estimate the prevailing wind direction because cloud streaks orient parallel to the prevailing winds.

### 3.1.4 North America and South America categories

The category from North America occurred more in winter or fall, associated with the cold fronts general circulation, as HYSPLIT trajectories suggested. We linked the South America category with the broad circulation of cyclones north of Puerto Rico promoting southerly low-level winds. We excluded the North and South America category because they only occurred a few times.

### 3.1.5 Exceptions to the classification method

The classification criteria (Table 1) is objective, and a subjective assessment of the classification results led to changes in the assigned classes a few times. Volcanic aerosols reach CPR if the Soufriere Hills' volcano emits simultaneously with southeast low-level winds, cloud streaks and trajectories. One exception is when the ash is already in the Atlantic northeast of Puerto Rico (Figure 3). In this instance, winds shifting to the northeast can bring volcanic ash to Puerto Rico.

Also, we used SAL images cautiously because of the existence of specific cases that MODIS suggested heavy dust (Figure 4a) over Puerto Rico but SAL images suggested no dust (Figure 4b). Also, Figures 5a and 5b show cloud streaks oriented from northeast to southwest instead of southeast to northwest but we classified as AD because AD was clear. This example suggests that African dust can be transported from the Atlantic to CPR.

## 3.2 Extensive variables by aerosol category
### 3.2.1 Scattering ($\sigma_{sp}$) and absorption ($\sigma_{ap}$) coefficients

Figure 5a shows the $\sigma_{sp}$ frequency distributions for AD and for VA. We verified that these distributions are log-normal by taking the logarithm of the data and applying a normality test. For AD, the greater number of measurements ranged from ~10 to 100 Mm$^{-1}$ with fewer extremes showing gaps between them. Statistically, the gaps in the AD frequency distribution suggest that the more extreme AD events might have distinct causes. But studying the potential causes was out of our scope. Alternatively, the extreme events might all have the same cause, but the study period was insufficient to fill in the gaps. Figure 5a shows that AD had a greater range in the strength of $\sigma_{sp}$ (230 Mm$^{-1}$) in contrast to that of 75 Mm$^{-1}$ for VA.

Table 3 summarizes aerosol data collected from 2005-2010. Comparing the aerosol loads among aerosol categories within the MBL, we found that mean $\sigma_{sp}$ for AD was significantly greater than mean $\sigma_{sp}$ otherwise (3.5 times greater than CM, 1.9 times greater than CMS with greater sea salt content and 1.7 times greater than VA). Hence, mean





atmospheric aerosol load increased, on average, 3.5 times near the surface and the MBL if AD replaced CM. The greater number of AD events ranged from 30 to 35 Mm$^{-1}$ and regarding the VA case it ranged from 15 to 20 Mm$^{-1}$. Mean $\sigma_{sp}$ between AD and VA differed by 23 Mm$^{-1}$ with a 95% confidence interval (CI) from 16.8 to 27.8 Mm$^{-1}$. These results support our hypothesis about aerosol loads. Figure 5b shows the frequency distribution for CM and for CMS. The

coefficients of variation (the standard deviation over the mean) were smaller in the marine aerosols with 21% for CM and 22% for CMS. The coefficient of variation for AD was 65% and for VA (51%). Thus, AD and VA had coefficients of variation 3.1 and 2.4 times greater than those of the marine aerosols, respectively.

About the columnar data, Figure 6a shows a VA event for which Figure 6b shows the corresponding volume size
distribution dominated by fine aerosols (i.e. not the marine aerosols). Many studies regarding volcanic ash particles indicate the presence of coarse mode particles inside the plume. Due to gravitational settling and meteorological conditions the coarser particles fall earlier, while the finer travel longer distances. Thus, the aging of such aerosols affects the sizes of the detected VA particles. Trajectories of 1-3 days allowed coarse particles to fall and fine particles to reach Puerto Rico (Mather et al., 2003). Also, dome-forming eruptions of highly crystalline magma, such as that at
the Soufriere volcano in Montserrat, release much fine ash (Baxter et al., 1999; Moore et al., 2002; Bonadonna et al., 2002; Bonadonna et al., 2005). In contrast, for the VA event shown in Figure 6c, the volume size distribution (Figure 6d) shows coarse aerosols (such as the marine aerosols) dominating the volume. These results imply that VA events are variable enough that sometimes VA dominates the column loading and sometimes sea-salt dominates, implying that two distinct aerosols are present in the VA category. However, the optical data in Table 3 shows that VA dominates the light
scattering during VA events because the light scattering efficiency of coarse-mode particles is much lower than that of fine-mode particles. Therefore, in some VA events, sea-salt might dominate the column mass loading, but VA will dominate the scattering and optical depth.

Turning back to surface data, the ratio of the mean absorption coefficients ($\sigma_{ap}$) between AD and VA suggests that AD
absorbs 50% more sunlight than what VA absorbs. To calculate the local heating rate after the absorption by VA or by AD we combined the Beer's Lambert law for the absorption rate, $\frac{dF}{dz} = \frac{\sigma_{ap}}{\mu} F$ with the equation for the local heating rate, $\frac{dT}{dt} = \frac{1}{C_p \rho} \frac{dF}{dZ}$ and obtained, $\frac{dT}{dt} = \frac{\sigma_{ap}}{\mu C_p \rho} F$, where $F$ is the flux density at altitude $z$; $\frac{dF}{dz}$ is the absorption rate; $\mu$ is the cosine of the zenith angle; $\sigma_{ap}$ is the absorption coefficient; $C_p \rho$ is the heat capacity of air (1kJ K$^{-1}$ kg$^{-1}$) times $\rho$, the density of air; and dT/dt is the local heating rate. The equation for dT/dt shows that the heating rate is directly proportional to the
absorption coefficient. If we substitute the absorption coefficient for AD and for VA in the last equation, we get that the change in temperature because of AD is 50% greater than the change in temperature because of VA. This result only considers the change in temperature because of absorption but omits the effect on temperature because of radiation cooling. An implication of this result is the need to measure the absorption coefficient accurately. To keep or improve this accuracy needs that we continue improving the techniques to measure absorption. In the next section, we will analyze
the columnar optical depth, contrasting, or comparing it with the surface scattering.

### 3.2.2 Columnar Aerosol Optical Depth

Similar to the surface mean ($\sigma_{sp}$), the columnar mean (AOD) for AD was greater than that of other aerosols categories.
For example, the mean AOD for AD was 5 times greater than that of CM and 3.5 times greater than that of CMS. In



contrast, $\sigma_{sp}$ for AD was 3.5 times greater than that of CM and 1.9 times greater than that of CMS. The different ratios imply a greater AD fraction transported above, than within the MBL. Based on the SAL and on Reid et al. (2003a, b), African dust appears more often from the top of the MBL to the trade wind inversion up to 5000 m. Hence, the greater AD fraction above is due to the SAL that is transporting a greater amount of mass above the MBL. For the VA, mean

$\sigma_{sp}$ was 1.1 times greater than that for the CMS and the columnar mean AOD for the VA was 1.6 times greater than that for the CMS. Therefore, the VA enhanced the optical depth more than the surface scattering if the VA superimposed on the CMS. Also, the VA enhanced the scattering and increased the optical depth by a factor of two. Furthermore, mean $\sigma_{sp}$ for AD was 1.7 times greater than that for VA and mean AOD was 2.1 times greater. Therefore, within the MBL, AD increased the scattering 70% more than that by VA and doubled the columnar optical depth produced by VA.

### 3.3 Intensive variables by aerosol category

### 3.3.1 Columnar Aerosol Optical Depth Fine Fraction (AODFF)

Concerning particle sizes causing extinction in the atmospheric column, an overall mean AODFF of 0.27 implies that,

on average, coarse particles caused three-fourths of the visible light extinction by aerosols. In addition, mean AODFF among aerosol categories was significantly different, except between CM and CMS. This result suggests that in Puerto Rico, different aerosol categories have significantly different columnar size distributions. For instance, the mean AODFF in VA was 0.09 more than mean AODFF in AD with a 95% CI from 0.04-0.14. Hence, at $p<0.05$, columnar VA were on average, significantly smaller than columnar AD and columnar marine aerosols. In other words, we are more than

95% certain that columnar VA are smaller than columnar AD or marine aerosols. The mean AODFF in marine and AD aerosols differed significantly by ~0.03. These results support our hypothesis concerning aerosol sizes. Users of these data should decide if these differences, although statistically significant, are meaningful to them. Our results, in agreement with Reid et al. (2003b), show that coarse aerosols dominate the extinction of visible light in Puerto Rico.

### 3.3.2 Scattering ($R_{sp}$) and Absorption ($R_{ap}$) Fractions

Concerning the sizes of the scattering particles, mean sub-micron scattering fractions, $R_{sp}$, were on average low (Table 3), with a mean of 0.2. Hence, coarse aerosols produced 80% of the scattering within the MBL. In addition, mean $R_{sp}$ for VA was significantly greater than mean $R_{sp}$ for AD. Even though they only differed by 0.03 with a 95% CI from

0.01 to 0.05, the result means that we are more than 95% certain that, within the MBL, volcanic ash has a smaller fraction of coarse scattering particles than what AD and marine aerosols have. The lowest mean $R_{sp}$ in marine aerosols implies that marine aerosols have the greatest fraction (> 85%) of coarse aerosols. Therefore, the scattering particles sizes among natural aerosols in Puerto Rico were significantly different, supporting our hypothesis.

The overall mean $R_{ap}$ of ~ 0.5 showed that on average, the amount of coarse/fine absorbing aerosols in Puerto Rico, is similar within the MBL. The smaller mean $R_{ap}$ (0.37) for AD contrasts with the mean $R_{ap}$, otherwise, that only differed few hundredths from a mean of 0.6. Hence, coarse absorbing particles in AD produced 63% of the absorption. Therefore, if the absorption by fine AD particles was due to soot, or the AD fine mode, or a mix of these two, the result implies that absorption by the coarse iron oxides surpassed the absorption by the fine. A question arising is: Why are absorbing

aerosols smaller in VA than in AD if both are iron oxides? One explanation is that VA are generated by a mechanism in which aerosols are heated to elevated temperatures. These results also show that coarse aerosols produced most of the



extinction in Puerto Rico within the MBL and support the importance of the SAL as a transport mechanism allowing coarse aerosols to move from the Sahara to the Caribbean and farther to Miami (Prospero, 1999), after their lift to high altitudes. Our methods to segregate the sub-micron particles from the total allowed us to measure what fraction of the total scattering and absorption was due to sub-micron aerosols.

A surface mean $R_{sp}$ of 0.2 and a columnar mean AODFF of 0.27 agree that coarse aerosols produced most of the scattering within the MBL and most of the columnar extinction in Puerto Rico. But the overall mean $R_{sp}$ was 35% smaller than the columnar mean AODFF, implying that the size distributions at the surface vs. column average are a major contribution to this difference. Another contributor is the difference in the sizes separating the columnar fine/coarse

AOD and surface submicron/super-micron scattering or absorption. The size cut for surface data is 1 m aerodynamic diameter corresponding to 0.7 μm geometric diameter for a particle density of 2.0. The AERONET "size cut" is poorly defined but is larger than 0.7 μm geometric diameter. Also, the difference in relative humidity aloft vs. at the surface may have to do with the difference between $R_{sp}$ and the AODFF. The nonsignificant difference between the measured AOD at 500 nm and the interpolated AOD to 550 nm suggests that comparing/contrasting $\sigma_{sp}$ at 550 nm and AOD at

500 nm have a similar difference as comparing/contrasting them at the same wavelength (550 nm).

### 3.3.3 Single Scattering Albedo ($\omega_0$)

The difference between mean $\omega_0$ for AD and for VA was nonsignificant, even though the values of $\sigma_{sp}$ and $\sigma_{ap}$ to derive

$\omega_0$, were significantly greater for AD. Also, the absorbing particles were significantly larger for AD than the absorbing particles for VA. The mean $\omega_0$ for marine aerosols was ~0.99. We calculated the uncertainty in $\sigma_{ap}$ associated with the calibration constants in Eqn. (1) ($\sigma_{meas} = K_1\sigma_{sp} + K_2\sigma_{ap}$) of Bond et al. (1999), including the adjustment by Ogren (2010), with $\sigma_{meas}$, the apparent absorption. In this adjustment, $K_2 = 1.44 \pm 0.24$ and $K_1 = 0.02 \pm 0.02$ such as in Bond et al. (1999). With $a$ defined as $a = \frac{\omega_0}{1-\omega_0} = \frac{\sigma_{sp}}{\sigma_{ap}}$ and the relative uncertainty as $\frac{\sigma_{ap,cal}}{\sigma_{ap}} = \frac{((a*\Delta K_1)^2 + (\Delta K_2)^2)^{1/2}}{(a*K_1+K_2)^2}$, Eqn. (S4c) in

Sherman et al. (2015) becomes, $\frac{\sigma_{ap,cal}}{\sigma_{ap}} = \frac{\left(\left(0.02*\frac{\omega_0}{1-\omega_0}\right)^2 +(0.24)^2\right)^{1/2}}{\left(0.02*\frac{\omega_0}{1-\omega_0}+1.44\right)^2} = 17\%$ relative uncertainty for marine aerosols. The

term $\sigma_{ap,cal}$ is the uncertainty in $\sigma_{ap}$ resulting from uncertainties in the calibration constants and $\sigma_{ap}$ is the absorption coefficient. Mean $\omega_0$ was 0.96 for AD and for VA, with a relative uncertainty of 15%. The uncertainty in $\sigma_{meas}$ also contributes to the total uncertainty of $\Delta\sigma_{ap}/\sigma_{ap}$ and $\Delta\omega_0/\omega_0$. Uncertainties greater than 0.01 in $\Delta\omega_0/\omega_0$ would yield an $\omega_0$ greater than one, which is physically impossible. From these results, the marine aerosols absorption coefficient could be

zero. If the 95% CI for $\omega_0$ includes values greater than 1.0, we cannot reject the hypothesis that marine aerosols are non-absorbing. Even though values of $\omega_0$ above 1.0 are physically impossible, the measurement techniques, which are difference-based, means that measured values above 1.0 are possible. To analyze whether $\omega_0$ depends on å, we plotted $\omega_0$ vs. å in Figure 7 that shows a weak correlation, which may have to do with absorbing particles sizes not only in the fine mode but also in the coarse mode. Also, results for $R_{ap}$ showed that coarse AD aerosols had greatest absorption

coefficient. Smaller $\omega_0$ and å in Figure 7 suggests absorption by coarse aerosols such as the iron oxides in AD (Moosmüller et al. (2009). Our results for $\omega_0$ in AD (0.96) support the findings by E. A. Reid (2003) that elemental iron composes ~2.5-3% of the total dust mass.




### 3.3.4 Radiative Forcing Efficiency (ΔRF/AOD)

The mean radiative forcing efficiency (ΔRF/AOD) was significantly different among aerosol categories. For instance, mean ΔRF/AOD for AD was greater (about 2 W m$^{-2}$) than that of CM, and ~ 1 W m$^{-2}$ greater than those of CMS and VA. To compare the radiative cooling because of AD and because of VA, we first calculated the average of ΔRF/AOD

and AOD for the period and then found the product of the two averages for AD and for VA. We obtained a mean radiative cooling of -11.7 W m$^{-2}$ for AD and -5.7 W m$^{-2}$ for VA. Equivalently, AD produced around twice the cooling of VA at the top of the atmosphere, despite that AD produced a greater local heating rate due to absorption. Examining how ΔRF/AOD varies with the aerosol load, we plotted ΔRF/AOD vs. $\sigma_{sp}$ (Figure 8a). Figure 8a shows that ΔRF/AOD varies more over the first 5 - 45 Mm$^{-1}$. In section 3.3.1 we found a log-normal distribution for $\sigma_{sp}$. Therefore, one

explanation for the large variability of ΔRF/AOD is that the log-normal, skewed to the right distribution of $\sigma_{sp}$, creates a non-linear relation between $\sigma_{sp}$ and ΔRF/AOD. To make the relation linear, we transformed $\sigma_{sp}$ in $\log(\sigma_{sp})$, where $\log(\sigma_{sp})$ is the natural logarithm of $\sigma_{sp}$. Figure 8b shows a linear-log regression model of the form, $\frac{\Delta RF}{AOD} = \alpha + \beta \log \sigma_{sp}$ with $\alpha$ = -30.7 W m$^{-2}$ being the intercept on the vertical axis and $\beta$ = 1.15, the slope of the line. The model predicts that an increase of $\log\sigma_{sp}$ by one unit, changes ΔRF/AOD by 1.15. Substituting the smallest measured scattering coefficient

$\sigma_{sp}$ of around 10 Mm$^{-1}$ in the regression model gives $\frac{\Delta RF}{AOD}$ ~ -28 W m$^{-2}$. An increase of 20% in $\sigma_{sp}$ = 10 Mm$^{-1}$ would result in $\frac{\Delta RF}{AOD}$ = -30.7 W m$^{-2}$ + 1.15 log (1.2*10) W m$^{-2}$ = -28 W m$^{-2}$ + 0.21 W m$^{-2}$. Therefore, an increase of 20% in $\sigma_{sp}$ would result in 0.21 W m$^{-2}$ increase of ΔRF/AOD. We thus showed that greater loadings of AD aerosols caused greater radiational cooling at the top of the atmosphere and greatest local heating within the MBL and that ΔRF/AOD depends on the aerosol loading. We also estimated the increase in ΔRF/AOD as $\sigma_{sp}$ increase by a certain percent. From Table 3

we can determine the relative frequency of each aerosol category and how much each category contributes to ΔRF/AOD over Puerto Rico: AD contributes 58%, CM plus CMS 27%, and VA 15%.

Eqn. 6 shows that ΔRF/AOD also depends on two aerosol properties, single-scattering albedo ($\omega_0$) and upscatter fraction ($\beta$). To diagnose the overall relation in Figure 8a, we plotted the relation between ΔRF/AOD and these two aerosol variables separately (Figure 9 and Figure 10). The results show a clear dependence of ΔRF/AOD on the upscatter

fraction, illustrating the importance of knowing the backscatter fraction to calculate it and the importance of the aerosols' sizes. A clear dependence of ΔRF/AOD on $\omega_0$ is not seen in Figure 10. This is due to the fact that, in Puerto Rico, there is not enough variability in $\omega_0$. Summarizing, we found that besides the effects of the upscatter fraction and the single scattering albedo, the aerosol loading accounted for around 50% of the change in the radiative forcing efficiency.

### 3.4 Monthly climatology

Note that mean $\sigma_{sp}$ and AOD peaks in summer because of AD (Figure 11 (a) and (b)). The monthly variation of $\sigma_{sp}$ peaks in June and the mean variation of AOD peaks in July. Two explanations for the difference in timing of the maxima are: 1) we recorded $\sigma_{sp}$ at controlled RH and the AERONET assessed AOD at ambient RH and 2), the transport aloft brings different amounts of aerosols than the transport in the MBL. The result is consistent with Reid et al. (2002) that reported

highest dust concentrations in the MBL in June. The climatology of $\sigma_{sp}$ and AOD also agrees with Prospero (1999) that the seasonality for AD peaks in summer.

Mean $\sigma_{ap}$ was smaller in July than in other months (Figure 12) implying that most of the absorbing aerosols are above the MBL as suggested by the greatest columnar AOD. The maximum $\sigma_{ap}$ in May 8, 2008 was due to an AD event.





Because on May 7, 2008 we sensed VA at CPR and $\sigma_{ap}$ is extensive and additive, this high value may have to do with VA, although VA, if any, was underneath the AD. The absorption increased from 0.8 Mm$^{-1}$ on May 6, to 6 Mm$^{-1}$ on May 7 and 10.8 Mm$^{-1}$ on May 8. The corresponding $\sigma_{sp}$ increased from 65 to 165 to 219 Mm$^{-1}$ for the same days. The maximum $\sigma_{ap}$ in May 8, 2008 could be due to the $\sigma_{ap}$ of AD added to the $\sigma_{ap}$ of VA because $\sigma_{ap}$ is extensive and extensive
properties are additive.

Black carbon (BC) from forest fires in Brazil may have to do with the greater mean $\sigma_{ap}$ in September and October. We base this statement on the trajectories, forest fires in Brazil, and the southern long-range transport due to cyclones over the Atlantic. Absorption of 6 to 7 Mm$^{-1}$ had corresponding $\sigma_{sp}$ of 35 to 37 Mm$^{-1}$. Hence, greater absorption in September
and October may have to do with BC from the forest fires in SA. We saw this transport three days in October, and two days in September.

The low mean $R_{sp}$ in Figure 13 (a) shows a small variation through the year, suggesting that coarse marine aerosols are always present in Puerto Rico. The lower monthly mean $R_{sp}$ in June and in July suggests more coarse AD aerosols
arriving to the MBL enhancing the coarse marine aerosols. From March to May, and in September and October, a greater mean $R_{sp}$ suggests a smaller fraction of coarse particles. The $R_{sp}$ monthly variation at the surface resembles the monthly variation of the columnar AODFF (Figures 13 (a) and (b)). Namely, smaller mean $R_{sp}$ and AODFF in July, suggest more coarse mode scattering particles within the MBL and columnar, in July. Due to the continuous release of coarse sea-salt particles at the surface, the mean monthly $R_{sp}$ was lower than that of AODFF through the year. The two parameters have
low values showing that coarse aerosols (such as marine) dominated the scattering of within the MBL and extinction in the atmospheric column. The AD coarse mode enhanced the marine coarse mode in summer as shown by the lower values $R_{sp}$ and AODFF.

Absorbing particles were mostly sub-micron from January to May and in December (Figure 14). Even though the dust
season peaks in summer, the easterlies shape the local weather until November, suggesting that coarse AD aerosols dominated and caused a low mean $R_{ap}$ from June to November within the MBL. This result agrees with Prospero et al. (1970) and Prospero and Carlson (1972) that synoptic outbreaks of Saharan dust occur from late spring to fall. Volcanic ash caused the greatest mean $R_{ap}$ of around 0.7 and a maximum $R_{ap}$ of 0.95 on March 24, 2008.

Figure 15 (a) shows the monthly climatology of $\omega_0$. We have noted that September had the smallest mean $\omega_0$ and the median $\omega_0$ throughout the year stayed greater than 0.96. December, January, February, and March had mean monthly $\omega_0$ of ~0.97. This result implies that marine aerosols are always present and are thus important for the radiative balance. The lowest mean $\omega_0$ occurred in September with a mean of 0.95 and minimum about 0.90. The monthly variability of the radiative forcing efficiency ($\Delta RF/AOD$) throughout the year was greater in March, in April, and in May (Figure 15
(b)). The variability in the mean $\Delta RF/AOD$ was small through from 2006 – 2010. The larger range in these months may be due to the collective effects of the fine, absorbing particles of volcanic ash and the coarse, less absorbing marine aerosol. Monthly means, however, varied little ($27 \pm 1$ W m$^{-2}$).

**4 Conclusions**


We have analyzed the means and variability of the radiative-forcing properties of natural aerosols in Puerto Rico, and we have suggested several criteria to classify them according to their source. Mean loads, sizes, and absorbing properties




were different among aerosols in Puerto Rico at $p<0.05$. The radiative forcing properties were different not only within the MBL, but also in the atmospheric column. Therefore, we accepted our hypothesis for the parameters associated with the loads and with the sizes at the surface and columnar. We rejected the hypothesis for differences in the single scattering albedo between African dust and volcanic ash. However, the parameters computed with the single scattering albedo and the parameters used to compute the single scattering albedo were different.

The absorbing particles in African dust were larger than the absorbing particles in volcanic ash. Even though smaller sizes as in volcanic ash have greater scattering efficiency, African dust produced a greater radiative cooling at the top of the atmosphere due to it greater load. African dust could produce a local heating rate 50% greater than that of volcanic ash because of greater absorption coefficient. However, we could not distinguish among aerosols of distinct categories only knowing their means and variability because their frequency distributions overlap much.

Despite the uncertainty, our method to classify aerosols can, on average, contribute to understand the means and the variability in the radiative-forcing properties. Agreement between our results and the empirical evidence cited from earlier studies allow us to conclude that our classification method is feasible. We also presented the monthly climatology of aerosols radiative-forcing properties in Puerto Rico. Size-resolved measurements showed that the coarse mode dominated the scattering and absorption in African dust. Coarse marine aerosols dominated the volume (and mass) because they are near the source at the surface, despite falling faster than AD aerosols with a smaller size mode. Trajectories (Table 1) showed that AD aerosols last ~5-7 days to reach CPR. The continuous production at the surface makes marine aerosols important because they are a continuous source of scattering. Continuing efforts to measure or estimate upscatter fraction, such as in Andrews et al. (2006) to estimate the asymmetry factor, would help to decrease the uncertainty in calculated radiative forcing efficiency, this is because our results showed that variations in upscatter fraction strongly contribute to variations in radiative forcing efficiency. Because different absorbing aerosols cause different changes in temperature, the accurate measurement of absorption continues as one key issue in the study of the climate-forcing properties of aerosols.

In this work the PSAP values were adjusted to 550 nm, but we have used the AOD values measured at 500 nm. We have done some interpolations to analyze the data at 550 nm. Our preliminary results indicate that the AOD values interpolated to 550 nm are not significantly different from the measured values at 500 nm. Therefore, we believe that the use of the values at 500 nm fall within the scope of the present work. Nonetheless it remains our intention to take up the case of AOD values adjusted at 550 nm in more details in further work in the near future.





*Acknowledgements*. We gratefully acknowledge Patrick Sheridan (NOAA), and Anne Jefferson and Derek Hageman (University of Colorado) for their contributions to station operations and data processing, as well as the NOAA Climate Program Office for the funding support. We also acknowledge the NSF AGS 0936879 and the NSF GK-12 Scholarship for their financial support, and the Conservation Trust of Puerto Rico for the use of their facilities at the nature reserve

5   of Cabezas de San Juan. (CSJ). Thanks to the Atmospheric Chemistry and Aerosols Research Group at UPR-RP for their support (especially to Mr. Félix Zürcher, Ian Gutiérrez and Carmelo Costacamps). We thank Gerardo Morell for his financial support through NASA EPSCOR and José F. Nieves for their financial support through Department of Physics PEAFs.

30



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

.



**Table 1. Summary of the criteria to classify natural atmospheric aerosols in Puerto Rico.**

Aerosol categories

| *Criteria* | CM | CMS | VA | AD | NA | SA | Unknown |
|---|---|---|---|---|---|---|---|
| $\sigma_{sp}$ | < 20 Mm$^{-1}$ | > 20 Mm$^{-1}$ | N/A | N/A | N/A | N/A | N/A |
| $\sigma_{ap}$ | < 0.6 Mm$^{-1}$ | < 0.6 Mm$^{-1}$ | N/A | N/A | N/A | N/A | N/A |
| AOD | < 0.1 | > 0.1 | N/A | N/A | N/A | N/A | N/A |
| HYSPLIT BT at 100, 500, 1000 m, AGL at 06 and 12Z | Not over land 3-4 days before arrival to PR | Not over land 3-4 days before arrival to PR | SE or across VA areas last 1-3 days | E-SE last 5-7 days | NW last 4-7 days | SW-SSE | N/A |
| Cloud streaks (if present) | WNW-ENE small | WNW-ENE elongated | SE-NW | E-SE | N/A | N/A | |
| MODIS Terra/Aqua | Dark blue ocean color with a few or no white caps. | Dark blue ocean with larger number of white caps | Show VA emissions | Dust observed leaving Africa 5-7 days before dust observed over CPR. Brown color. No VA observed. | N/A | N/A | Overcast skies observed or lack of MODIS images |
| SAL images | No dust suggested over CPR | No dust suggested over CPR | No dust suggested over CPR | Dust suggested over CPR | No dust suggested over CPR | No dust suggested over CPR | N/A |

CM is clean marine, CMS is clean marine with greater sea salt content, VA is volcanic ash, AD is African dust, NA is North America, SA is South America. Unknown means the category could not be defined.



**Table 2. Flowchart to complement Table 1**



**Table 3. Aerosol optical properties by aerosol category. Overall measurement results for means ± statistical standard deviation. Values in parenthesis are the number of daily averages. The left column says if the variable is extensive, intensive, at the surface or columnar. All calculated values are for λ=550 nm except $\mathring{a}$ and $\sigma_{ap}$, calculated for the 550/700 nm wavelength pair. AOD was assessed at 500 nm.**

| Aerosol category | | | | | |
|---|---|---|---|---|---|
| **1. Extensive** | CM | CMS | VA | AD | ALL |
| a. surface | | | | | |
| $\sigma_{sp}$ (Mm$^{-1}$) | 15.6 ± 3.21 (110) | 27.3 ± 6.10 (109) | 30.9 ± 15.9 (150) | 52.8 ± 34.2 (362) | 33.4 ± 26.7 (731) |
| $\sigma_{ap}$ (Mm$^{-1}$) | 0.51 ± 0.17 (32) | 0.54 ± 0.22 (26) | 1.26 ± 1.20 (80) | 1.83 ± 1.51 (324) | 1.6 ± 1.5 (462) |
| b. columnar | | | | | |
| AOD | 0.09 ± 0.03 (41) | 0.13 ± 0.031 (39) | 0.21 ± 0.061 (60) | 0.45 ± 0.17 (179) | 0.28 ± 0.15 (319) |
| **2. Intensive** | | | | | |
| a. columnar | | | | | |
| AODFF | 0.21 ± 0.091 (41) | 0.19 ± 0.11 (39) | 0.35 ± 0.16 (60) | 0.26 ± 0.15 (179) | 0.27 ± 0.16 (319) |
| b. surface | | | | | |
| $R_{sp}$ | 0.15 ± 0.04 (110) | 0.14 ± 0.05 (101) | 0.26 ± 0.11 (150) | 0.23 ± 0.09 (358) | 0.22 ±0.10 (719) |
| $R_{ap}$ | 0.62 ± 0.14 (51) | 0.63 ± 0.17 (23) | 0.65 ± 0.11 (64) | 0.39 ± 0.18 (209) | 0.48 ± 0.24 (347) |
| $\mathring{a}$ | 0.33 ± 0.16 (110) | 0.33 ± 0.18 (101) | 0.63 ± 0.49 (150) | 0.44 ± 0.34 (358) | 0.46 ± 0.36 (719) |
| $\omega_0$ | 0.99 ± 0.01 (76) | 0.99 ± 0.01 (70) | 0.96 ± 0.03 (82) | 0.96 ± 0.02 (330) | 0.96 ± 0.03 (558) |
| $\Delta RF/AOD$ (Wm$^{-2}$) | -28 ± 0.061 (76) | -27 ± 0.75 (70) | -27 ± 1.1 (82) | -26 ± 0.87 (331) | -26.7 ± 1.25 (559) |


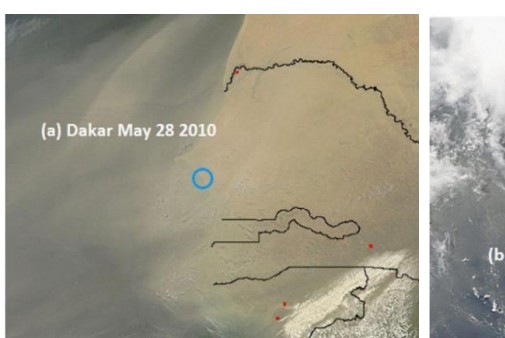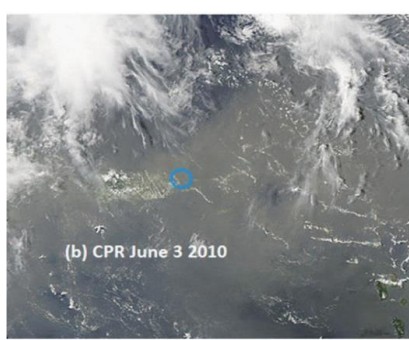

Figure 1. (a) African dust cloud over Dakar, Africa. (b) African dust cloud over Puerto Rico, the Caribbean and Atlantic. (Source: https://aeronet.gsfc.nasa.gov/cgi-bin/bamgomas_interactive)

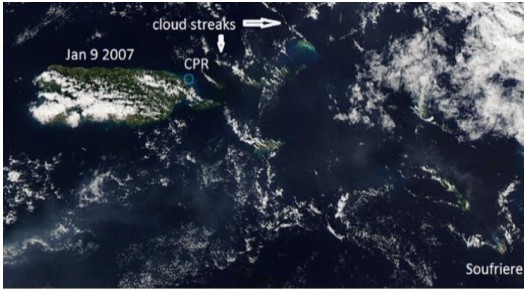

Figure 2. Volcanic ash reaching Puerto Rico under a southeast flow.
(Source: https://aeronet.gsfc.nasa.gov/cgi-bin/bamgomas_interactive)

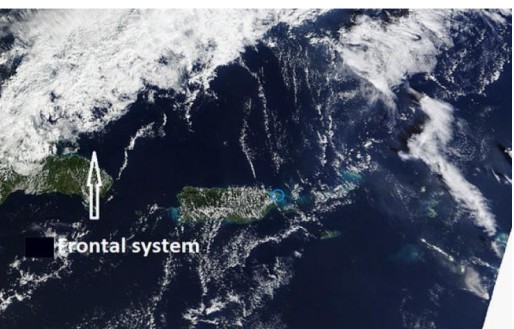

Figure 3. Volcanic ash plume northeast of Puerto Rico.
(Source: https://aeronet.gsfc.nasa.gov/cgi-bin/bamgomas_interactive)




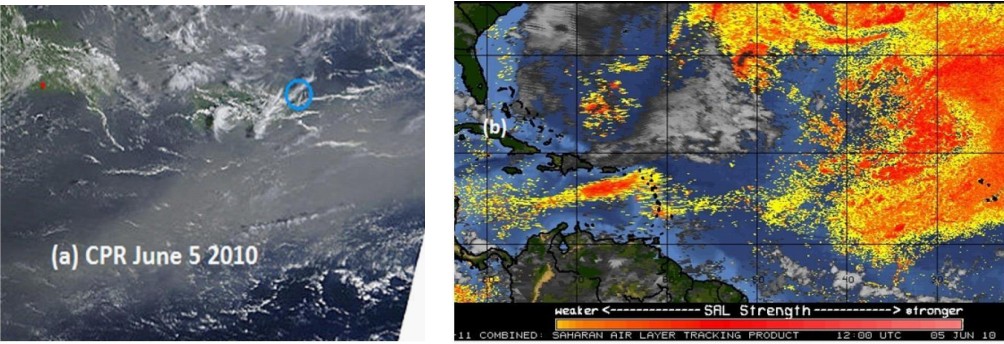

Figure 4. (a) Dust cloud over Puerto Rico and adjacent waters June 5, 2010. (b) SAL image June 5, 2010.
(Source: (a) https://aeronet.gsfc.nasa.gov/cgi-bin/bamgomas_interactive, (b) http://tropic.ssec.wisc.edu/archive/).

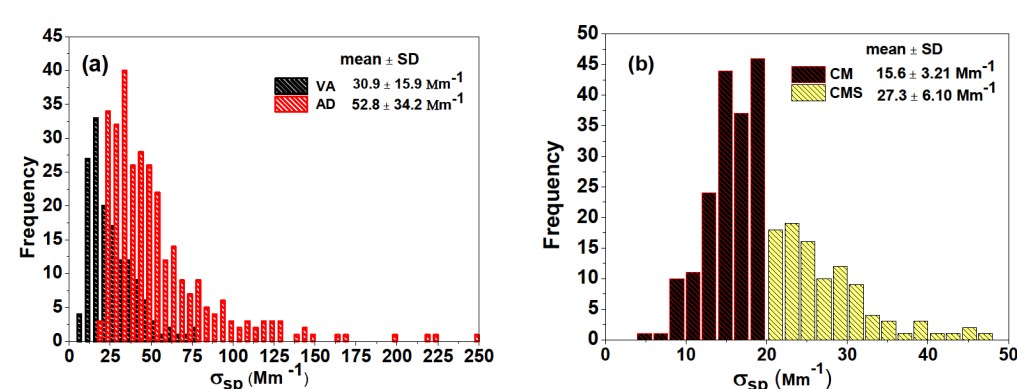

Figure 5. (a) Frequency distribution of the scattering coefficient $\sigma_{sp}$, for AD and for VA. (b) for CM and for CMS.



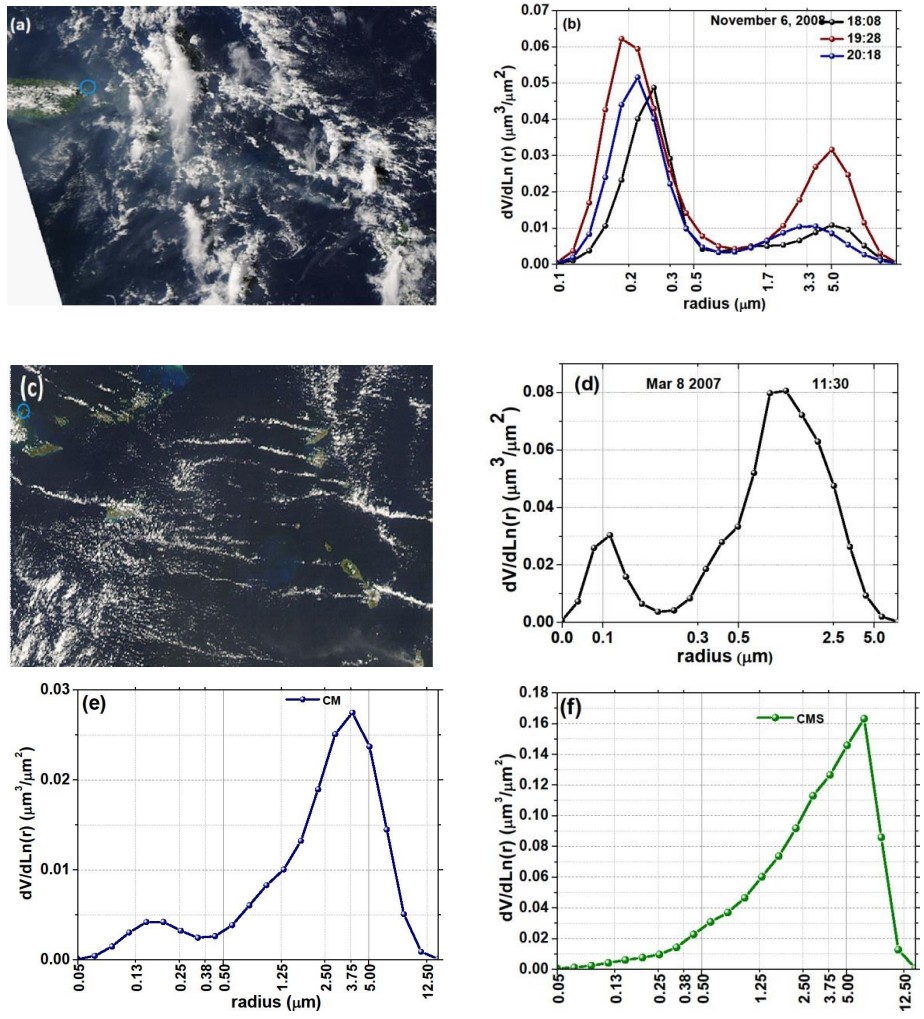

Figure 6. (a) Volcanic ash event over CPR in November 6, 2008. (b) Volume size distribution at various times in November 6, 2008. (c) Volcanic ash event on March 8, 2007. (d) Volume size distribution corresponding to the event on March 8, 2007. (e) Mean volume size distribution for the CM category. (f) Mean volume size distribution for the CMS category.  (Source: Figures 7 (a) and (c)  https://aeronet.gsfc.nasa.gov/cgi-bin/bamgomas_interactive)





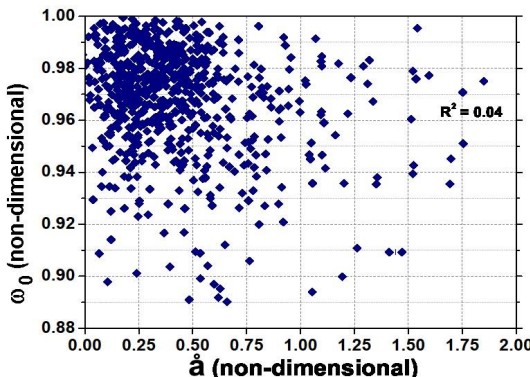

Figure 7. Variation of the single scattering albedo with the Ångström exponent (å).

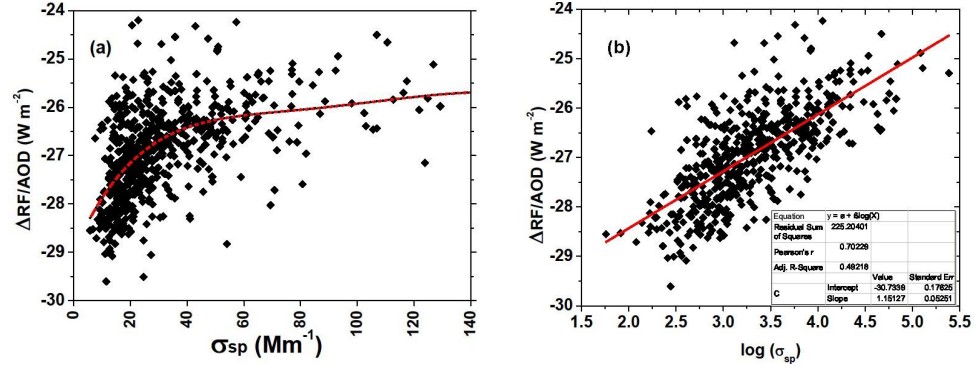

Figure 8. (a) Variability of the radiative forcing efficiency $\Delta RF/AOD$ with the scattering coefficient $\sigma_{sp}$. (b) Variability of the radiative forcing efficiency $\Delta RF/AOD$ with the natural logarithm of $\sigma_{sp}$.

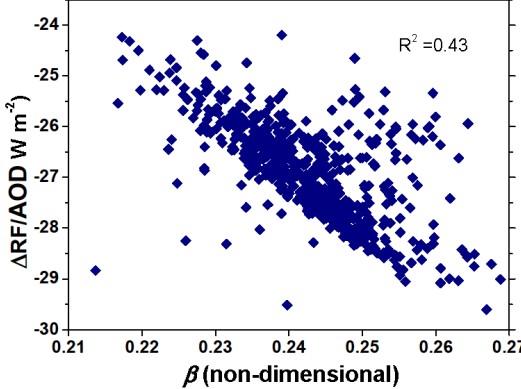

Figure 9. Variability of $\Delta RF/AOD$ with the size dependent $\beta$.





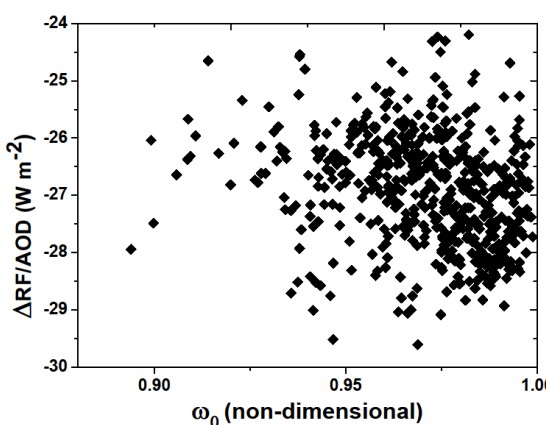

Figure 10. Variability of the Radiative forcing efficiency ΔRF/AOD as a function of the single scattering albedo ($\omega_0$).

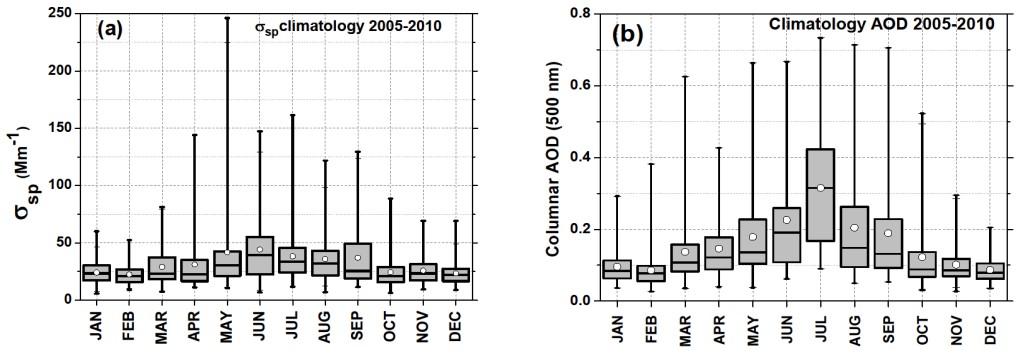

5    Figure 11. (a) Climatology of the scattering coefficient ($\sigma_{sp}$). (b) Climatology of the columnar optical depth (AOD) using daily averages.

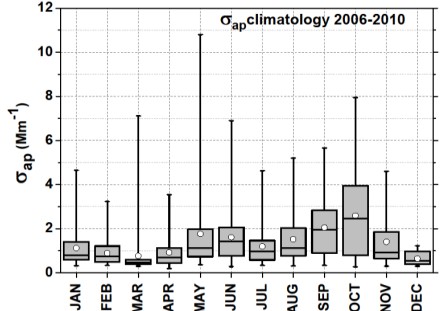

10    Figure 12. **Climatology** of $\sigma_{ap}$ using daily averages.





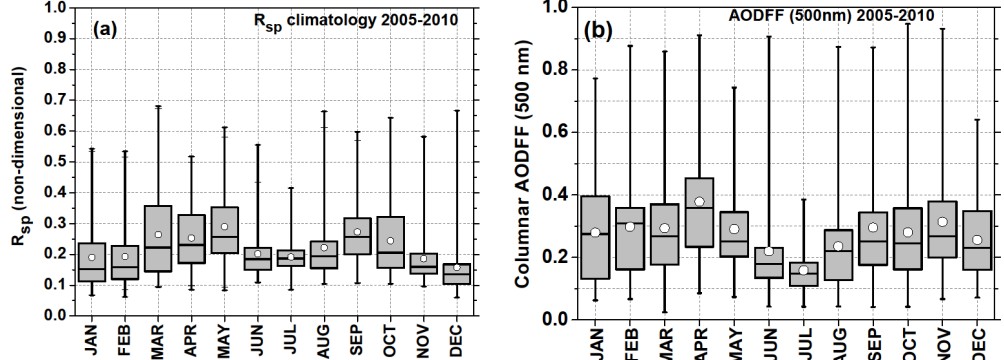

Figure 13. (a) Climatology of $R_{sp}$. (b) Climatology of columnar aerosol optical depth fine fraction AODFF, using daily averages.

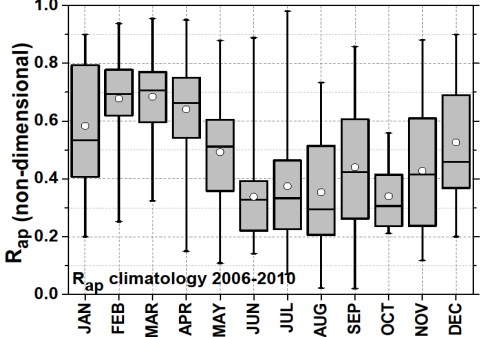

Figure 14. Climatology of $R_{ap}$ using daily averages.

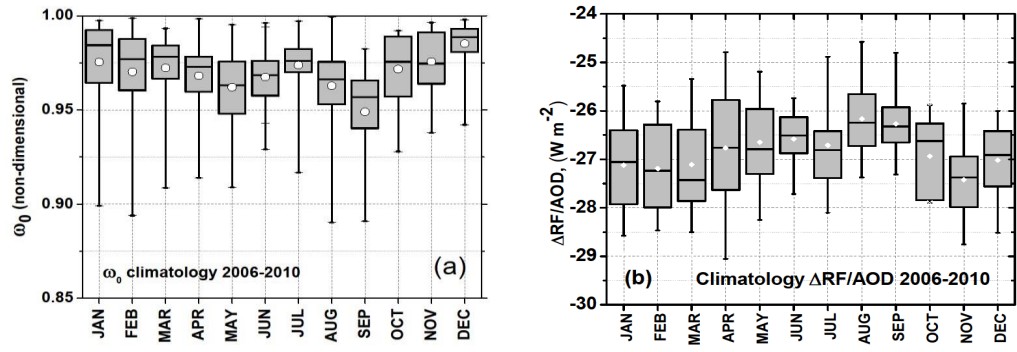

Figure 15. (a) Climatology of $\omega_0$ and (b) Climatology of $\Delta RF/AOD$ using daily averages.