# Peer review of "Variations in the physicochemical and optical properties of natural aerosols in Puerto Rico - Implications for climate"

_Atmospheric Chemistry and Physics, 2018_

## Referee Comment (RC1) · Anonymous Referee #1 · 6 Sep 2018

General Comments:

The authors are utilizing observations of aerosol properties, performed both at surface and in the atmospheric column, which are further classified by the air masses arriving over the Cape San Juan Atmospheric Observatory in Puerto Rico, in order to roughly estimate the aerosol impact on the direct radiative forcing. The manuscript is well written with a good scientific sound and thus to my opinion worth being published in the Atmospheric Chemistry and Physics journal. However, in order to be improved I kindly suggest to the authors to take into consideration the following comments.

Specific Comments:

[Figure]

1. Page 2, line 2: "because of their difference". Maybe "because of their great variability" is more suitable.

2. Page 2, line 25: "space-time" instead of "pace-time"

3. Page 6, line 6: "impossible values". I think that "and data with no physical meaning" is more appropriate.

4. Page 11, line 4: Consider deleting the second "because" due to redundancy.

5. Page 11, line 12: Since the letters $\alpha$ and $\beta$ are also used when referring to aerosol properties (also within the manuscript), for not misleading try to avoid using them, and instead just replace them with x and y (or with any other letter that is not used for referring to something else).

6. Page 11, line 33: I agree with this explanation however, you have to excess the 50-60% of RH in order to enhance the scattering efficiency by a factor of around 1.5. Do these RH levels occur during July over the site of observation?

7. Page 12, line 10: Please mention to what "SA" refers to.

8. Page 12, line 10: Is it possible that only 2-3 observations of BC (?) can affect so much your 4 years of statistics? I have some doubts on this, which are getting stronger when looking that the statistically mean and median values of absorption which are almost the same. This is true for both months of September and October.

9. Table 2 is not a table but a Figure. In this figure try to be consistent and use either "Y" & "N" or "Yes" & "No". Also provide the threshold value missing in scattering efficiency of CM.

10. In Table 1 please correct VA refers to Volcanic Aerosol and not to Volcanic Ash. This is the secondary sulfate portion, associated to the fine mode fraction of the volcanic plume. Is this the reason behind the relatively high values (compared to the ones observed for AD) of Angstrom exponent and single scattering albedo presented for VA

in Table 2?

---

## Referee Comment (RC2) · Anonymous Referee #2 · 12 Sep 2018

The overarching goal of this work is to characterize aerosol radiative properties and aerosol radiative forcing efficiencies for three major aerosol types over Puerto Rico. This would provide observational constraints to improve Radiative Forcing predictions due to aerosol-radiation interactions (RFari) from Chemical Transport Model (CTM). Uncertainties in CTM spatio-temporal distributions of speciated aerosol have significant implications in the CTM-estimated aerosol Direct Radiative Forcing, especially through propagating uncertainties in CTM species-dependent RFari. We strongly encourage the authors to work on the major revisions below and resubmit their work, as the result would lead to substantial contribution to scientific progress within the scope of this journal.

[Figure]

1. The overall motivation of this study should be clearly tied to the improvements of RFari in models

2. The method to define each aerosol type should be more clearly described. MODIS, SAL and VAAC need to be clearly defined with references. A flow chart combining the information of Table 1 and Fig. 2 would help illustrate the method. Fig. 2 needs to be corrected and is incomplete. For example, it is missing information on AOD and sigma_ap thresholds as well as MODIS RGB image visual inspection in the case of CM types. Another example is an infinite loop where sigma_sp <20 can be no and sigma_sp>20 can also be no; one arrow out of the CM-related trajectories box needs to say no instead of yes. The method in section 3.1 should, for example, explain the reasoning behind looking at similar directions for low level winds and cloud streaks. Similarly, the method behind defining exceptions to the classification method (section 3.1.5) is not clear enough.

3. The authors should discuss the sources of errors in their aerosol type classification; many of their choices are arbitrary and need more justification.

4. The authors should reference other aerosol type classification methods e.g. Russell et al., 2014, Patadia et al., 2013; Lee et al., 2016

5. The description of the location of each instrument, winds, volcano etc. deserves a map in the opening of section 2

6. In comparing AOD and scattering coefficients, the authors often omit to discuss the effects of the absorption coefficient and very frequently directly compare both measurements. Comparing these two measurements might not make much sense as they are taken at the surface or full-column measurements and the in-situ measurements are not humidified, compared to ambient AERONET-AOD measurements.

7. It is not clear why the authors separate intensive and extensive properties in their analysis; and some extensive properties are mentioned in the intensive properties section which makes it confusing.

8. The authors need to discuss the limitations and error sources on their RF efficiency calculations.

9. Throughout the paper, among other objectives, the authors aim to test a particular hypothesis, which is only very briefly described in the introduction i.e., "means and variability of aerosol from different sources differ significantly at $p<0.05$". This hypothesis needs to be clearly defined and deserves at least a paragraph including a few references.

10. A few scientific terms are used in a less conventional way throughout the paper such as aerosol "load" (usually describes aerosol concentration), or "radiative forcing properties" (instead of usually saying "aerosol radiative properties" or "aerosol radiative forcing").

11. The description of the dF/dz calculation should be in the method section

12. The AERONET aerosol mode definition (what the authors refer to "size cut") is defined as the inflexion point in the AERONET-derived size distribution.

13. Many equations (such as in section 3.2.1 or 3.3.3), if kept in the text, deserve to be numbered

14. The purpose of Fig. 7 is not clear. It shows the extinction angstrom exponent (EAE) versus single scattering albedo (SSA) for all aerosol types. Russel et al. [2014] (among other papers) show a separation of different aerosol types in the same EAE-SSA space. It would be interesting to show something similar using the CPR-related results per aerosol types and quantify the separation among each cluster of points.

15. The authors should consider showing the results of Fig 11 through 15 for each aerosol type.

Lee, H., Kalashnikova, O. V., Suzuki, K., Braverman, A., Garay, M. J., and Kahn, R. A.:

limatology of the aerosol optical depth by components from the Multi-angle Imaging SpectroRadiometer (MISR) and chemistry transport models, Atmos. Chem. Phys., 16, 6627-6640, https://doi.org/10.5194/acp-16-6627-2016, 2016. Patadia et al..: Aerosol airmass type mapping over the Urban Mexico City region from space-based multi-angle imaging, Atmos. Chem. Phys., 13, 9525-9541, 10.5194/acp-13-9525-2013, 2013. Russell P. B., M. Kacenelenbogen, J. Livingston, O. Hasekamp, S. Burton, G. Schuster, M. Johnson, K. Knobelspiesse, J. Redemann, S. Ramchandran, and B. N. Holben, A Multi- Parameter Aerosol Classification Method and its Application to Retrievals from Spaceborne Polarimetry, Paper #: 2013JD021411R, J. Geophys. Res. Atmos., accepted June 27, 2014.
* * *